# A Study on the Characteristics of Sports Athletes’ YouTube Channels and User Reactions

**DOI:** 10.3390/bs14080700

**Published:** 2024-08-12

**Authors:** Bora Moon, Taeyeon Oh

**Affiliations:** 1Department of Sports Science, Sungkyunkwan University, Suwon 16419, Republic of Korea; bora505@skku.edu; 2Seoul AI School, aSSIST University, Seoul 03767, Republic of Korea

**Keywords:** YouTube, contents, athlete, characteristics, reactions

## Abstract

This study examined the content characteristics and user responses of athlete-run sports YouTube channels, providing empirical insights for content production strategies and contributing to the development of athlete-run sports YouTube channels. Content analysis was conducted on 3306 videos posted on 20 popular YouTube channels of South Korean athletes from 1 January 2020 to 31 December 2021. The formal characteristics analyzed included video length, the presence of foreign language subtitles, paid advertisements, and information sources. The content characteristics examined were the types of sports events, main content themes, and whether the content matched the athlete’s sport. Results revealed significant differences in content characteristics and user responses based on whether the athletes were active or retired. This study’s distinctive contribution lies in highlighting the evolving role of athletes as content creators and providing strategic implications for enhancing the competitiveness of athlete-run sports YouTube channels. Future research should consider a broader range of sports YouTubers and a wider variety of YouTube channels to gain comprehensive insights into the sports content ecosystem on this platform.

## 1. Introduction

The advent of digital technology has significantly transformed the production and consumption of content [1]. The traditional media content industry has shifted to an open structure, allowing individuals to freely produce and share content, with individual preferences and choices becoming paramount, thus fostering active content utilization [2]. YouTube is at the forefront of this transformation. As the most widely used video-sharing and social media platform globally [3], YouTube boasts approximately 1 billion monthly users, with around 500 h of video uploaded and over 1 billion hours of video watched daily [4]. Unlike other social media platforms, YouTube has a revenue-generating model that allocates advertising revenue to channels with more than 1000 subscribers and 4000 h of viewing time within the past 12 months. This model has given rise to professional YouTubers, creating new employment opportunities [5,6]. A YouTuber is defined as an individual who regularly produces and shares original content on YouTube [7]. According to a 2022 survey, the recognition of YouTuber as a career is notably high in South Korea (hereafter, Korea), where it ranks as the third most popular profession among teenagers’ career aspirations [8]. Indeed, Korea leads the world in the number of YouTubers and in YouTube viewing and content production relative to its population [9].

This study contributes to the literature on digital media, social media communication, and sports communication by examining athlete-run sports YouTube channels. Sport-related channels are increasingly active on YouTube, leading to the emergence of star YouTubers with fan bases exceeding one million individuals. Recently, a growing number of active and retired athletes have launched their own YouTube channels, engaging in the platform as content creators. Historically, sports YouTube channels managed by non-athletes, such as “Shoot for Love” and “Ball in Love”, were more prevalent, with athletes primarily appearing as guests. However, there has been a significant shift, with athletes now managing their own YouTube channels to directly interact with fans [10]. This evolution has transformed athletes from being merely consumers or guests on sports YouTube channels to becoming active content producers. Notably, the sports genre demands specific expertise, and users recognize that athlete YouTubers inherently possess this expertise, distinguishing athlete-run sports YouTube channels from others [11]. Therefore, a systematic review is warranted to examine the format and content produced on athlete-run sports YouTube channels and the audience’s reactions to them.

As the media environment evolves with the advancement of digital technology, we are entering an era where individuals can freely produce and share a wide range of content on social media. YouTube is at the forefront of this shift, driving the online video consumption market by generating diverse content and fostering a unique consumption culture [12]. In Korea, 91.5% of PC and mobile Internet users utilize YouTube for watching online videos [13]. As of September 2022, the average monthly viewing time was 30 h and 34 min, significantly surpassing the global average of 23 h and 24 min. Additionally, Korea boasts the highest number of YouTubers per capita worldwide [14]. This indicates that Korea leads globally in both content production and YouTube viewing participation rates.

A YouTuber is defined as an individual who regularly produces and shares original content on YouTube in this study. The term encompasses both those who consider YouTubing their primary occupation and those who engage in it to earn supplementary income. This distinction is important as it reflects the varying degrees of professional engagement and economic reliance on the platform. In this study, the focus includes both active and retired athletes who have embraced YouTubing, either as a primary profession or as an additional endeavor, contributing to the rich diversity of content on the platform.

The prominence of sports content has grown alongside the expansion of the YouTube platform [15]. Among Korea’s top 100 YouTube channels, genres such as music entertainment, sports, games, education, and know-how/style are predominant [2]. Sports content ranks fourth among the 17 genres, constituting approximately 10% of all content posted on YouTube [16]. According to a survey by the Korea Communications Commission [17], YouTube holds the highest usage rate among online video service providers, with 22.7% of all online videos viewed being sports-related. This highlights the high level of interest in sports content in Korea and the substantial consumption of such content through YouTube. Furthermore, data from NoxInfluence, a YouTube data analysis company, reveals that as of January 2024, Korea’s leading sports channel had 6.08 million subscribers and 6.57 billion cumulative video views.

According to Chan-Olmsted [18], sports fans consume sports content using various platforms, and the production and consumption patterns of sports content differ depending on the media platform.

In traditional TV broadcasting, sports content typically includes broadcast videos, highlights, and news, each following common production practices. However, YouTube sports content is markedly more diverse [19]. This diversity arises from YouTube’s freedom from the programming regulations of traditional media, allowing anyone to create content, leading to an infinite variety of content types [20]. 

According to Moon [21], who analyzed the characteristics of sports YouTube channels, 83.9% of sports channels are operated by individuals, making it the most common type. Additionally, it was found that individually operated sports channels gain popularity by producing and sharing content that deviates from traditional sports content production practices and attracts the interest and consumption of users. Channels featuring various types of sports content—such as game broadcasts and highlight videos, game analysis, skill improvement lessons for recreational sports participants, home training sessions, and reviews influencing purchasing decisions for sports-related goods, clothing, and equipment—are steadily increasing [22]. Consequently, sports content users can watch less popular sports or amateur game videos that were not available on traditional media by searching on YouTube. They actively consume diverse content of their interested sports through the YouTube platform [23]. In this way, various sports’ content is being produced through YouTube, and users are actively consuming it.

As YouTube has evolved into a core platform for sports content, not only individual creators but also established sports broadcasters, companies, professional sports clubs, and associations are producing and distributing sports content on YouTube. Additionally, athletes have begun operating their own YouTube channels. Many retired professional Korean athletes, including Kim Byung-ji (former national soccer player), Ha Seung-jin (Former NBA player), Kim Dong-hyun (former UFC fighter), and Lee Hyung-taek (former professional tennis player), are now popular YouTubers. This trend indicates that as sports-related professions expand, athletes can choose to become sports YouTubers as a second career after retirement. To support this transition, the Korean Professional Sports Association has introduced the “Media Creator Introductory Program” to train retired athletes as YouTubers [24]. Recently, the phenomenon of athletes entering YouTube has extended to include active athletes. During the COVID-19 pandemic, when various professional sports were suspended or postponed, athletes who could not compete in public stadiums actively communicated with fans through YouTube [25].

Especially since sports is a genre that requires expertise, users’ expectations and reactions to YouTube channels operated by athletes, whose physical skills and careers are recognized, can differ from those operated by non-athletes [11]. According to Lou [26], athletes use video platforms like YouTube as tools to express themselves and reinforce their identities. Additionally, athletes use social media as a means of communication with their fans, generating significant revenue in the process [27]. Kim, Kim, and Kim [28] argue that sports fans want to access information about their favorite athletes and share their daily lives through social media platforms like YouTube, highlighting that the essential motivation for using social media is the intrinsic fandom of sports fans.

Thus, with the growth and spread of YouTube, the media industry’s focus has shifted from traditional media platforms to YouTube, causing various changes in the sports content market. In particular, YouTube content is being produced by leveraging the expertise and experience of athletes, and content is actively consumed based on fandom. Therefore, a systematic review of the types and content of videos produced on athletes’ YouTube channels and the users’ reactions to them is necessary.

As the media environment evolves, the types and forms of sports content on YouTube have diversified, with content being produced in ways distinct from traditional media to promote consumption [29,30]. According to Stuaff [15], factors such as screen composition, editing skills, scene selection, commentators’ public confidence, and commentary expertise have a greater influence on viewing satisfaction in YouTube sports content than in traditional sports media. This indicates that the characteristics of sports content are more significant on YouTube compared to other existing sports media. In response, YouTube content producers strategically plan and apply the most effective video production elements to attract user interest, maximize views, and secure a large subscriber base from the planning stage [19].

YouTubers establish relationships with their subscribers through unique communication methods, with content being central to maintaining these relationships [31]. The process of consuming content and achieving high view counts on YouTube involves deciding on the content and form, considering various factors that influence consumer selection [32]. YouTubers shape their identity with consumers through their choice of video genre and production format [33]. Formal characteristics of YouTube content—such as production type, subtitle service, paid advertisements, content length, and information source—also significantly impact user responses [16,34,35,36,37].

On YouTube, an informant refers to a person or speaker who appears directly in the content and provides information to the audience [36]. The effectiveness of messages with identical content can vary depending on the informant, making informants crucial in content production and consumption [38]. Research has long shown that highly reliable informants are more persuasive and lead to more positive behavioral and attitudinal changes [39,40]. Online users tend to favor content produced by informants with high reliability and expertise in their fields, making them more likely to share and disseminate the content and information online [41,42,43]. With the diversification of informants, their roles in online media, including YouTube, are gradually increasing [44]. Additionally, user reactions vary depending on the informant featured in YouTube content [34,37]. 

Expertise is a key characteristic of YouTube sources [45], and when users perceive a source to be highly knowledgeable, they respond positively to the information provided [46]. In particular, according to Kim [47], fitness-related subscribers are more attracted to reliable creators and professional information than to general fitness and health information. In this context, it is expected that the presence of expert sources on sports athletes’ YouTube channels will lead to more positive user responses.

The length of YouTube content is another crucial characteristic affecting user responses. Influenced by the “snack culture”, content produced as short videos, approximately 15 min in length, is gaining popularity among consumers [48]. YouTube content often features video formats that cater to this snack culture. Cheng, Liu, and Dale [16] reported that 98% of YouTube content is under 600 s, with romantic videos averaging 200 s and sports videos around 230 s being particularly popular. Joo [35] also demonstrated that content length has a statistically significant impact on view counts, with formal characteristics such as production methods influencing the number of views. Content characteristics play a critical role in user reactions and their selection process [32]. Waters and Jones [49] highlighted that choosing a channel’s topic is essential for content curation, significantly impacting the channel’s mid- to long-term identity and scalability on YouTube. Consequently, differences in view counts can be observed depending on the topic of the YouTube content [35,50].

In the realm of sports content, events can serve as themes representing content characteristics, with view counts varying according to different sports events [19]. Additionally, whether the primary content of an athlete-run YouTube channel aligns with the athlete’s specific sport can influence user reactions. As previously noted, higher reliability and expertise of the information source lead to more favorable user responses to the information provided by YouTubers [42,43]. Therefore, it is expected that users will respond positively to sports event content, particularly when the reliability and expertise of the information source are highly regarded.

In response, this study aims to examine the characteristics and user reactions to the content of sports YouTube channels operated by active and retired athletes. Specifically, this research seeks to address the following guiding questions: (1) What are the key characteristics and formats of the content produced on athlete-run sports YouTube channels? (2) How do user reactions vary based on these content characteristics and the athlete’s status (active vs. retired)? This comprehensive examination provides insights into the unique attributes of athlete-run sports YouTube content and offers strategic implications for enhancing the competitiveness of these channels, thus holding both academic and industrial significance.

By addressing these guiding questions, the study aims to offer a comprehensive understanding of the dynamics of athlete-run sports YouTube channels. Specifically, examining the key characteristics and formats of these channels is essential for identifying common practices and industry standards, enabling comparisons with other types of sports content on YouTube and determining elements contributing to success. Investigating how user reactions vary based on content characteristics and the athlete’s status provides insights into audience preferences and engagement, crucial for optimizing videos for user satisfaction and higher engagement rates. Understanding whether content characteristics differ between active and retired athletes can help tailor content strategies to the unique needs and capabilities of each group, given their different motivations and resources.

Exploring these aspects sheds light on the types of content that attract and retain viewers, offering a deeper understanding of what makes certain content popular. Analyzing differences in user responses based on content characteristics allows content creators to identify which types of content are more effective in engaging audiences and fostering loyalty. Understanding if and how content characteristics differ between active and retired athletes can provide insights into how an athlete’s status influences their content creation and audience engagement strategies.

Assessing user reactions to the content on athlete-run sports YouTube channels is vital for evaluating the success and impact of these channels, helping to understand audience sentiment and overall content reception. Collectively, these inquiries aim to offer strategic implications for enhancing the competitiveness and effectiveness of athlete-run sports YouTube channels, thus holding both academic and industrial significance.

## 2. Method

This study employed content analysis to examine the characteristics of athlete-run YouTube channels and their corresponding user responses. Content analysis is a research method that objectively, systematically, and quantitatively describes explicit communication content [51]. It is used to objectively and systematically analyze the characteristics of a message and make inferences [52]. Given its suitability for examining communication content, content analysis was deemed the most appropriate method for achieving the objectives of this study.

The subsequent sections detail the analytic targets, data collection procedures, coding categories, reliability measurements, data processing methods among coders, and considerations of research ethics.

### 2.1. Analysis Object

In this study, videos posted on 20 popular YouTube channels of former and current Korean athletes from 1 January 2020 to 31 December 2021 were selected for analysis. The number of views and subscribers serve as objective indicators of YouTuber awareness and loyalty [53], making them the main criteria for channel selection [54,55]. Accordingly, the top 20 most popular channels were selected by comprehensively considering the number of subscribers and views using NoxInfluence, a YouTube channel analysis site. NoxInfluence provides objective information on YouTube channels, including the number of subscribers, views, and rankings, and has been used in several previous studies [6,55,56,57].

For this study, ten YouTube channels of former athletes in Korea, ranked highest by the number of subscribers and views as of 1 January 2022, were selected. Additionally, ten YouTube channels of current athletes were included in the analysis. A total of 3306 videos were analyzed, with the analysis unit being videos posted over the past two years from 1 January 2020 to 31 December 2021, on the selected 20 popular channels.

### 2.2. Data Collection

Data were collected by three coders, including a researcher and two graduate students majoring in the sociology of sport. Training coders in content analysis is crucial for enhancing the reliability of the analysis [58]. To ensure consistency and accuracy, a content analysis protocol was first established for data collection. The three coders were then trained in the categorization of each nominal coding category. Following their training, they conducted the data collection, with continuous confirmation and correction performed throughout the coding process to maintain accuracy and reliability.

### 2.3. Coding Categories

The coding categories in this study are broadly composed of the formal characteristics, content characteristics, and user reactions to athlete-run sports YouTube channels. Building on the existing literature on sports media and sports YouTube channels, the coding categories were refined to align with the study’s purpose and objectives through a consultation with two experts in sports media. Table 1 provides a detailed breakdown of the specific contents of the coding categories.

First, the analysis items were revised and supplemented based on previous studies to understand the formal characteristics of athlete-run sports YouTube channels [16,28,35,37,59,60]. Formal characteristics of the content were classified into video length, presence/absence of subtitles, presence/absence of paid advertisements, and information sources. For detailed analysis, video length was categorized into four groups: less than 3 min, 3–10 min, 10–20 min, and 20 min or more. Videos were also categorized based on the presence or absence of subtitles. Similarly, they were divided into those with and without paid advertisements. Additionally, videos were classified based on the presence of the informant alone or the informant with others, such as members of the general public, active and retired sports players, sports-related experts, celebrities, and others.

The content characteristics were used to examine the specifics of the athlete-run YouTube channels and were also revised and supplemented based on previous studies [19,20,35,50,61]. These characteristics were classified based on the sport type, main content, and whether the main content matched the athletes’ primary sports. Videos were categorized by the sport mainly covered, including baseball, soccer, basketball, volleyball, golf, boxing, tennis, mixed martial arts, skating, fitness, and other sports. The main content categories included overseas sports event analysis, domestic sports event analysis, lessons and information, content about a specific player or club, training, entertainment, daily sharing, eating shows (mukbang), and product advertisements. Following previous studies, user reactions were measured by the number of views, likes, and comments [28,62].

### 2.4. Inter-Coder Reliability and Data Analysis

In this study, 520 videos (approximately 16% of the total analysis targets) were analyzed to measure inter-coder reliability, which was assessed using the Holsti Index. A Holsti Index score of 80% or higher indicates that the analysis results are considered reliable [52]. The inter-coder reliability analysis yielded a Holsti Index of 0.84, confirming the reliability of the analysis. The collected data were analyzed using the statistical software SPSS for Windows version 20.0. As a result of the normality test conducted on user responses, which are the dependent variables of this study, it was found that the data did not satisfy normality. When data do not meet normality, non-parametric statistical analysis is performed [62,63,64]. The Mann–Whitney U test is a non-parametric statistical technique that compares the ranks of data included in two groups, and the Kruskal–Wallis test compares the ranks of data included in three or more groups to identify differences between groups [65,66].

Frequency and chi-square analysis were conducted, along with the Mann–Whitney U and Kruskal–Wallis tests. This study was approved by the Bioethics Committee of the Korea Institute of Science and Technology (approval number: KH2022-114).

## 3. Results

### 3.1. Formal Characteristics of Athlete YouTube Channel Content

The formal characteristics of athletes’ YouTube channel content were analyzed. Table 2 presents the results of the analysis.

The average video length for the sample was 624 s, and 1998 videos were 3–10 min in length, accounting for 60.4%. This was followed by 981 cases (29.7%) of 10–20 min, 185 cases (5.9%) of ≥20 min, and 142 cases (4.3%) of <3 min. Only 570 (17.3%) of the videos had foreign subtitles. There were 2736 cases (82.8%) with only Korean-language subtitles. A total of 664 videos (20.1%) contained paid advertisements, while 2842 videos (79.9%) did not. Regarding major sources, the number of videos featuring athletes running YouTube channels was 1670, accounting for 50.5% of the videos. Of these, 648 (19.6%) were active, 279 (8.4%) were retired, 277 (8.4%) were ordinary athletes, 250 (7.6%) were sport experts, 173 (5.3%) were celebrities, and 9 (0.3%) were others.

### 3.2. Differences in User Response Depending on the Formal Characteristics of Athlete-Run YouTube Channel Content

The Mann–Whitney U test and Kruskal–Wallis test were conducted to examine whether there were differences in user responses depending on the formal characteristics of athlete-run YouTube channel content. These non-parametric methods are appropriate because they do not assume a normal distribution of the data and are suitable for comparing differences between groups with ordinal or non-normally distributed interval data, which is common in user response metrics on social media platforms. Table 3 presents the results of this analysis. As Table 3 shows, there were statistically significant differences in user responses for all formal characteristics of the content.

### 3.3. Differences in the Formal Characteristics of YouTube Channel Content Depending on Athlete Status (Active vs. Retired)

Chi-square analysis was conducted to examine whether there is a difference in the formal characteristics of the content depending on whether the YouTube channel operator is active or retired. This method is appropriate because it is used to test the independence of categorical variables, allowing for the comparison of the distribution of formal content characteristics across the two distinct groups of active and retired YouTube channel operators. Table 4 presents the results of this analysis, which indicate statistically significant differences in all content format characteristics depending on whether the YouTube channel was operated by a former or current athlete.

### 3.4. Content Characteristics of Athlete-Run YouTube Channel Content

An analysis of the types of sport events mainly covered in the content of athlete-run YouTube channels revealed that content dealing with golf was the most frequent, accounting for 39.4% of the 1301 cases. There were a total of 694 cases of football (21.0%), 474 cases of content not related to sport (13.3%), 209 cases of mixed martial arts (6.3%), 144 cases of skating (4.4%), 134 cases of boxing (4.1%), 130 cases of basketball (3.9%), 103 cases of tennis (3.1%), 56 cases of fitness (1.3%), 44 cases of other sports (1.3%), 14 cases of volleyball (0.4%), and 4 cases of baseball (0.1%). Table 5 shows the results of the analysis of the content characteristics of athlete YouTube channel content.

Regarding the main content of athlete YouTube channels, lessons and information videos conveying skills or sport-related knowledge were the most frequent, accounting for 39.6% of the 1308 videos, followed by entertainment (661 cases or 20.0%), content about specific players or teams (637 cases or 19.3%), training (213 cases or 6.4%), daily life sharing (179 cases or 5.4%), product advertisements (80 cases or 2.4%), overseas sport game analysis (66 cases or 2.1%), eating show (mukbang) (61 cases or 1.8%), domestic sport game analysis (59 cases or 1.8%), and other (39 cases or 1.2%). Our analysis of whether the main content of the athletes who run the YouTube channels matched their sports revealed that 2697 videos, accounting for 81.6%, matched. Most were about sport events in which the athletes are or were engaged. Content unrelated to sport events appeared in 475 cases (14.4%), followed by 134 videos (4.1%) that mainly dealt with sports unrelated to the athletes’ main professions.

### 3.5. Differences in User Reactions Depending on the Content Characteristics of Athlete-Run YouTube Channel Content

The Kruskal–Wallis test was conducted to examine whether there were differences in user responses depending on the content characteristics of athlete-run YouTube channel content. This method is appropriate because it is a non-parametric test that allows for the comparison of more than two independent groups, making it suitable for analyzing user responses that are likely ordinal or non-normally distributed across different content characteristics. Table 6 presents the results of this analysis, which indicate a statistically significant difference in user responses for all content characteristics.

### 3.6. Differences in Content Characteristics of YouTube Channel Content Depending on Athlete Status (Active vs. Retired)

Chi-squared analysis was conducted to examine whether there is a difference in content characteristics depending on whether the YouTube channel operator is an active or retired player. Table 7 presents the results of the analysis which indicate statistically significant differences in all content characteristics depending on whether the YouTube channels are operated by current or former athletes.

### 3.7. User Reactions to Athlete-Run YouTube Channel Content

User reactions to the YouTube channel content of athletes were analyzed. Table 8 presents the results.

The average number of views of athlete-run YouTube channel content was 289,910.81. The lowest number of video views was 740, and the highest number of video views was 19,396,173. The average number of comments on athlete YouTube channel content was 344.83. There were also videos without comments, and the largest number of comments was 17,200. The average number of likes for the athlete YouTube channel content was 3937.87. The lowest number of likes was 15 and the highest number of likes was 278,400.

### 3.8. Differences in User Reactions Depending on Athlete Status (Active vs. Retired)

A Mann–Whitney U test was conducted to determine whether there was a difference in user responses depending on whether the YouTube channel operator was active or retired. As Table 9 shows, there is a statistically significant difference. The average ranking of current athlete-run YouTube content was significantly higher than that of retired players. As such, it can be seen that the average user response of active athlete YouTube channels is higher than that of retired athletes.

## 4. Discussion

The characteristics of the content and user reactions were analyzed by examining the YouTube channels of athletes. Differences were observed in content characteristics and user reactions between channels run by current and former athletes, as well as in user reactions based on the content characteristics. First, the formal characteristics and user reactions to athlete-run YouTube channel content were examined in detail.

The analysis of video length revealed that most videos lasted between 3 and 10 min. These results align with Cheng et al. [16], who found that more than 97.8% of sports content posted on YouTube lasted fewer than 600 s. The prevalence of snack culture, which involves enjoying short periods of leisure (approximately 10 min) while consuming snacks, has become a universal cultural phenomenon. YouTube is a representative platform where snack culture content is distributed and consumed [67]. Accordingly, it is believed that sports athletes are also producing videos that match this cultural trend. In particular, there was a notable difference in video length depending on whether the athletes were active or retired, with a higher proportion of shorter content (<10 min) found in the YouTube channels of current athletes. Generation Z (born in the mid-1990s to early 2000s), who are accustomed to watching videos on mobile devices, prefer short content that can be easily watched anytime and anywhere [68]. Consequently, this phenomenon is more pronounced in the YouTube content of relatively younger, current sports athletes.

However, an analysis of user reactions based on the length of content on athlete-run YouTube channels found that longer videos received relatively higher user reactions. This result may be attributed to the genre characteristics of sports content, which typically require a longer narrative to maintain the natural flow and tension of sports-based stories [69]. Cheng et al. [16] found that while comedy videos averaging 200 s and sports videos averaging 230 s are popular on YouTube, sports content generally requires a longer narrative. This could explain why relatively longer sports content remains popular, as it better accommodates the extended narrative structure inherent to the genre.

User reactions were higher when foreign language subtitles were provided, yet only 17.3% of all videos included foreign language captions. Given YouTube’s global reach [3], implementing strategies to provide foreign language captions could significantly enhance user engagement. S. Kim and E. Kim [70] noted that the top five most-viewed Korean YouTube videos in the beauty and makeup category all offered foreign language subtitles. Similarly, Byun [5] suggested that expanding YouTube content consumption globally by providing foreign language captions is essential for revitalizing YouTube channels.

A significant difference was observed in the provision of foreign language captions based on the athletes’ status (active vs. retired). Specifically, only 3.9% of retired athletes’ content included foreign captions, whereas 43.5% of active athletes’ content did. This difference may be due to some current athletes being active in overseas sports leagues, thereby catering to an international fan base. An analysis of comments on these channels shows many overseas fans leaving comments in their native languages. Sports fans often strengthen their attachment to sports through the information provided in YouTube content [71]. Therefore, it is crucial not only to expand the target audience by providing foreign captions but also to enhance communication with international fans.

YouTubers typically aim to generate high profits from their content [72]. Our analysis revealed that 20.1% of the videos contained paid advertisements, while 79.9% did not. With the growing popularity of YouTube content and the increasing public interest and trust in YouTubers [73], they are becoming significant marketing influencers, producing both videos and advertisements. Advertising revenue is a crucial motivator for YouTubers to continue creating content [74]. According to the Korea Creative Content Agency [20], advertisements accounted for the largest portion of individual media content creators’ income (59.3%) in 2021. However, while only 9.1% of active athletes’ YouTube content included paid advertisements, a significantly higher proportion (25.7%) of retired athletes’ content did. This higher percentage may be because retired athletes rely more on generating revenue through YouTube compared to active athletes, who also earn salaries from their sports teams. Full-time media creators tend to have higher average monthly revenues than part-time creators, with advertising being the largest revenue source [75].

However, users exposed to YouTube videos with advertisements tend to feel fatigue and a sense of intrusion, negatively impacting their immersion [76]. Consequently, the number of views and likes for content containing paid advertisements can be relatively low. Despite this, the number of comments on such content was higher, likely due to events offering products or vouchers through the comments section.

A major information source is another important formal characteristic of YouTube content [28,31,36,77]. With the general public increasingly producing and distributing information on YouTube, the role and proportion of public informants have gained focus [77]. However, on athlete-run YouTube channels, there is a high percentage of content led by athletes themselves, with active and retired athletes, as well as sports experts, acting as major informants. Statistically significant differences were observed in the main sources of information based on whether the YouTuber was an active or retired athlete. The proportion of active athletes providing major sources was relatively higher than that of their retired counterparts. This indicates that athletes tend to employ experts with skills and experience in sports as their main sources of information, leveraging their human networks. Both current and former athletes are likely to collaborate with other athletes and sports experts based on their personal connections and relationships.

The characteristics of athlete-run YouTube content, where the proportion of major information sources with expertise is high, are closely related to user reactions. Even for the same message, the effect varies depending on the information source [38]. Spence et al. [42] and Teng et al. [43] found that higher reliability and expertise of information sources on social media lead to more favorable user responses and a greater willingness to share the information. Similarly, this study found that content on athlete-run YouTube channels featuring celebrities and other retired and current athletes as main informants garnered high user responses.

We analyzed the content characteristics, specifically the type of sport event, main content, and whether the athletes’ content matched their respective sports. In the realm of sports content, an event serves as a theme or attribute representing the content’s characteristics [19]. Golf events accounted for the largest amount of content, with 1301 cases (39.4%). Public interest in golf has surged significantly in Korea, leading to the emergence of the term “golin” (novice golfer) and marking a second heyday for the golf industry [23]. This golf craze has resulted in a rapid increase in golf-related YouTube channels [78], particularly those run by active or retired professional golfers [79]. Therefore, it is unsurprising that most of the content dealt with golf events.

Differences in the types of sports events covered were observed based on the athletes’ status (active vs. retired), and there were statistically significant differences in user reactions depending on the sport. This finding aligns with J. Kim and K. Kim [19], who found that the number of views varied according to the type of sporting event. Different sports media content has distinct narrative structures, leading to varying types of viewing satisfaction for audiences [80]. Thus, user reactions differ depending on the events primarily covered in YouTube content.

An analysis of the main content of athlete-run YouTube channels revealed that lessons and information videos (39.6%) were the most common, conveying sports skills or sports-related knowledge to YouTube users. Searches for educational and instructional videos are prevalent on YouTube, which serves as an effective educational medium by enabling users to acquire information [57]. Unlike other media that convey information through text or images, YouTube uses videos to enhance consumer understanding of the content [81]. Consequently, YouTube is a convenient means of acquiring information, especially for younger generations more familiar with visual content. Following the COVID-19 pandemic, home learning has become popular, and many young people use YouTube as a primary method to learn specific sports and physical activities [82]. Thus, content featuring lessons and information was the most prominent, particularly on the YouTube channels of retired athletes.

User reactions varied according to the main content. Gupta, Singh, and Sinha [83] argue that the informational and entertainment aspects of YouTube content are linked to positive consumer reactions. Similarly, we found high user reactions to mukbang (eating shows), daily sharing, entertainment, training/lessons, and informational content. Notably, user responses to mukbang content were the highest. This finding aligns with Seol’s [2] observation that individual YouTubers gain popularity through mukbang-oriented entertainment content. In Korea, videos featuring YouTubers interacting with viewers while eating are becoming immensely popular [84]. Consequently, the user response to mukbang content was the highest on the YouTube channels of sports players.

Next, we analyzed whether the main contents covered on athlete-run YouTube channels matched the sports events in which the channel operators and athletes were involved. The results showed that 81.6% of the videos’ content aligned with the events of the athletes, indicating that most sports channel operators focus on the sports events in which they are or were involved. YouTube serves as an online space where people share, spread, and communicate their experiences, creating content that reflects their interests [85]. Therefore, athletes typically produce and share content based on their personal experiences.

Differences were observed in the alignment of main content and sports events on YouTube channels between active and retired athletes, which correlated closely with the main content analysis results. Retired athletes often produce lessons and information content related to their sports, leveraging their athletic careers, leading to a high match between main content and sports events. In contrast, current athletes frequently produce vlog-style content that entertains viewers with glimpses into their daily lives. Additionally, with the postponement of many professional sports league events due to COVID-19, the number of active professional athletes opening YouTube channels to share their daily lives and communicate with fans has increased [25]. As a result, the proportion of content not directly related to sports was relatively higher on the YouTube channels of current athletes.

This analysis revealed differences in the formal and content characteristics of YouTube channel content based on whether the athletes are active or retired, as well as statistically significant differences in user reactions. Sports fans form strong bonds by communicating with players and seeking news about their daily lives or information about their favorite athletes through social media [86]. Consequently, both offline and online spaces are becoming arenas for sports fandoms [78], with fandom activities particularly flourishing on online platforms like YouTube [87]. YouTube users can subscribe to their preferred channels, rate videos, leave comments, and express their opinions [88].

User responses to the YouTube content of active athletes were higher, as sports fans are more likely to subscribe to their channels, consume their content, and interact with them. Comparing the number of subscribers of the athletes analyzed in this study, the average number of subscribers to the YouTube channels of retired athletes was approximately 247,000, while the average number of subscribers to current athlete-run YouTube channels was approximately 336,000, indicating a significant difference. There is a correlation between the number of subscribers and the number of views [2], and a strong correlation exists between user response indicators such as views, comments, likes, and favorite registrations [89]. Therefore, the YouTube channels of active athletes with larger fan bases have more subscribers and relatively higher user reactions.

This study makes several significant academic contributions to the fields of digital content production, social media marketing, and sports communication by systematically analyzing the characteristics and user responses of athlete-run sports YouTube channels. First, the research expands the literature on digital content production and social media marketing. By exploring the unique attributes of athlete-produced content on YouTube, this study provides new insights into the strategies that athletes employ to engage with audiences on social media platforms. These findings contribute to a deeper understanding of how digital content production is evolving within the sports industry.

Second, the study highlights the evolving role of athletes as content creators. Traditionally seen as consumers or guests on sports media platforms, athletes are now increasingly becoming content producers themselves. This transformation underscores the dual role athletes now play, leveraging their sports expertise and fan base to create and share content. This shift not only enhances their engagement with fans but also provides a new avenue for personal brand building and revenue generation.

Third, the research offers strategic implications for enhancing the competitiveness of athlete-run sports YouTube channels. By identifying key content characteristics and their impact on user responses, the study provides practical recommendations for content creators. These insights can help athletes and other sports content producers optimize their YouTube content to better meet audience preferences and increase engagement.

Fourth, the study addresses a notable gap in the research on the sports content ecosystem on digital platforms. While much of the existing literature focuses on professional sports clubs and traditional media, this study centers on athlete-run YouTube channels, providing a fresh perspective on the production and consumption of sports content. The focus on South Korea, a leading country in YouTube usage and content production, further enriches the contextual understanding of this phenomenon.

Lastly, the methodological approach of conducting a content analysis on a large dataset of videos establishes a robust framework for future research. The systematic examination of video characteristics and user responses offers a replicable method that can be applied to similar studies in different contexts or with different subjects.

In conclusion, this study not only contributes to academic knowledge but also offers actionable insights for practitioners in the field of digital media and sports communication. Future research should build on these findings by including a broader range of sports YouTubers and exploring a wider variety of YouTube channels to gain more comprehensive insights into the dynamic and rapidly evolving sports content ecosystem on digital platforms.

## 5. Conclusions

YouTube is recognized as a familiar and attractive platform for the public, surpassing the influence of traditional global media companies. Consequently, a diverse array of sports content is being produced and consumed on YouTube, which has emerged as a core platform in the sports content industry. Recently, athletes who were once consumers or guests on sports YouTube channels have transformed into content producers and are actively working as YouTubers. YouTube is thus gaining attention as a new platform for athletes to engage with audiences.

Given the increasing activities of athlete YouTubers, this study aimed to provide a comprehensive understanding of sports YouTube channels by examining the characteristics of sports content produced on athlete-run YouTube channels and user responses. This analysis contributes to enhancing the competitiveness of athlete YouTube channels and broadening their reach.

However, this study has several limitations that should be explicitly acknowledged. First, the analysis was limited to the top 20 YouTube channels based on popularity rankings, which may not represent the full spectrum of athlete-run channels. This selective sampling could lead to bias, as popular channels may have different characteristics and user engagement patterns compared to less popular or emerging channels. Consequently, the findings may not be generalizable to all athlete-run YouTube channels.

Second, the study focused only on former and current athletes who have transitioned into YouTubers, potentially overlooking other influential sports content creators who may not have an athletic background. This narrow focus limits the ability to fully understand the diversity and dynamics of the sports content ecosystem on YouTube.

Third, the research did not account for the potential impact of regional differences, language barriers, or cultural factors that could influence content production and user responses on a global platform like YouTube. These factors could significantly affect the generalizability of the results to different contexts and demographics.

Future research should address these limitations by including a broader range of sports YouTubers, not limited to former athletes, to provide a more comprehensive analysis. Additionally, expanding the research to encompass a wider variety of YouTube channels related to sports, including those with varying levels of popularity and from different regions, would offer deeper insights into the sports content ecosystem on this platform. Moreover, longitudinal studies could be beneficial to examine how athlete-run YouTube channels evolve over time and their long-term impact on the sports media landscape.

## Figures and Tables

**Table 1 behavsci-14-00700-t001:** Coding categories of composition contents.

Coding Categories		Details
Formal characteristics	Video length	<3 min, 3–10 min, 10–20 min, >20 min
	Availability of foreign language subtitles	Foreign language subtitles provided/not provide
	Presence of paid advertising	With/without paid ads
	Informant	No information sources other than YouTubers, general public, retired athletes, current athletes, sports experts, celebrities, etc.
Content characteristics	Sport type	baseball, football, basketball, volleyball, golf, boxing, tennis, mixed martial arts, skating, fitness, other sports, not applicable to sports
	Main content	Overseas sport game analysis, domestic sport game analysis, lessons and information, content about specific athletes or teams, training, entertainment, daily life sharing, eating shows (mukbang), product advertisements, etc.
	Whether it matches the main content and the athlete’s sports	Match, non-match (other sport), content unrelated to sport
User reaction	Number of views	
	Number of likes	
	Number of comments	

**Table 2 behavsci-14-00700-t002:** Formal characteristics of athlete YouTube channel content.

Formal Characteristics	Frequency	%
Video length Average = 624 s, minimum = 12 s, maximum = 17,042 s		
<3 min	147	4.3
3–10 min	1998	60.4
10–20 min	981	29.7
>20 min	185	5.9
Availability of foreign language subtitles		
Foreign language subtitles provided	579	17.2
No foreign language subtitles	2736	82.8
Whether paid advertising is included		
Contains paid advertising	664	20.1
No paid advertising included	2642	79.9
Informant		
None other than the YouTuber	1670	50.5
General public	277	8.4
Former sports player	279	8.4
Active sports player	648	19.6
Sport expert	250	7.6
Celebrity	173	5.6
Others	9	0.3

**Table 3 behavsci-14-00700-t003:** Differences in user response according to formal characteristics of athlete YouTube channel content.

		Number of Views	Number of Comments	Number of Likes
Video length	χ^2^ (df)	146.09 (3)	200.51 (3)	153.23 (3)
	*p*	0.00	0.00	0.00
	Mean Rank			
	<3 min	888.17	1043.88	844.79
	3–10 min	1600.08	1529.46	1603.63
	10–20 min	1811.51	1917.33	1818.09
	>20 min	1980.05	2062.06	October 1940
Availability of foreign language subtitles	U	394,046.50	396,651.50	332,179.50
	*p*	0.00	0.00	0.00
	Mean Rank			
	Foreign language subtitles provided	2330.19	2325.62	2438.73
	Foreign language subtitles not provided	1512.52	1513.47	1489.91
Whether paid advertising is included	U	710,746.00	830,573.00	668,785.00
	*p*	0.00	0.03	0.00
	Mean Rank			
	Contains paid advertising	1402.90	1723.64	1339.71
	No paid advertising included	1716.48	1635.87	1732.36
Informant	χ^2^ (df)	284.75 (6)	179.48 (6)	247.01 (6)
	*p*	0.00	0.00	0.00
	Mean Rank			
	None other than the YouTuber	1573.66	1490.83	1691.93
	General public	1634.44	1552.42	1436.88
	Former athletes	1498.53	1732.99	1394.60
	Active athletes	2060.82	1962.52	1942.66
	Sport expert	1034.66	1600.55	970.06
	Celebrity	2105.33	2195.90	1975.75
	Others	1037.44	1279.06	1185.83

**Table 4 behavsci-14-00700-t004:** Differences in formal characteristics of content according to YouTube channels of retired and current athletes. Unit: cases (%).

		Active Player	Retired Player	Sum
Video length	<3 min	56 (5.0)	86 (3.9)	142 (4.3)
	3–10 min	720 (64.6)	1278 (58.3)	1998 (60.4)
	10–20 min	255 (22.9)	726 (33.1)	981 (29.7)
	>20 min	84 (7.5)	101 (4.6)	185 (5.6)
	χ^2^ = 44.37, df = 3, *p* < 0.001			
Availability of foreign language subtitles	Foreign language subtitles provided	485 (43.5)	85 (3.9)	570 (17.2)
	No foreign language subtitles	630 (56.5)	2106 (96.1)	2736 (82.8)
	χ^2^ = 812.86, df = 1, *p* < 0.001			
Presence of paid advertising	Contains paid advertising	102 (9.1)	562 (25.7)	664 (20.1)
	No paid advertising	1013 (90.9)	1629 (74.3)	2642 (79.9)
	χ^2^ = 125.37, df = 1, *p* < 0.001			
Informant	None other than the YouTuber	452 (40.5)	1218 (55.6)	1670 (50.5)
	General public	62 (5.6)	215 (9.8)	277 (8.4)
	Retired sports player	52 (4.7)	227 (10.4)	279 (8.4)
	Active sports player	402 (36.1)	246 (11.2)	648 (19.6)
	Sports expert	52 (4.7)	198 (9.0)	250 (7.6)
	Celebrity	88 (7.9)	85 (3.9)	173 (5.2)
	Others	7 (0.6)	2 (0.1)	9 (0.3)
	χ^2^ = 359.11, df = 6, *p* < 0.001			

**Table 5 behavsci-14-00700-t005:** Content characteristics of athlete-run YouTube channels.

Content Characteristics	Frequency	%
Sports type		
Baseball	4	0.1
Football	694	21.0
Basketball	130	3.9
Volleyball	13	0.4
Golf	1301	39.4
Boxing	134	4.1
Tennis	103	3.1
Mixed martial arts	209	6.3
Skating	144	4.4
Fitness	56	1.7
Other sports	44	1.3
Not a sport	474	13.3
Main content		
Overseas sport game analysis	66	2.1
Domestic sport competition analysis	59	1.8
Lessons and information	1308	39.6
Specific athletes and clubs	637	19.3
Training	213	6.4
Entertainment	661	20.0
Daily life sharing	179	5.4
Eating show (mukbang)	61	1.8
Product advertisement	80	2.4
Others	39	1.2
Whether the main content matches the athlete’s event		
Same	2697	81.6
Non-match (other event)	134	4.1
Content unrelated to sport	475	14.4

**Table 6 behavsci-14-00700-t006:** Differences in user response according to content characteristics of athlete-run YouTube channels.

		Number of Views	Number of Comments	Number of Likes
Sport type	χ^2^ (df)	602.265 (11)	1006.047 (11)	458.897 (11)
	*p*	0.00	0.00	0.00
	Mean Rank			
	Baseball	1115.00	1228.38	812.63
	Football	1357.55	1706.66	1279.51
	Basketball	2362.18	2427.12	2070.29
	Volleyball	3141.46	3173.04	3248.54
	Golf	1371.86	1113.57	1532.39
	Boxing	1720.15	1662.37	1399.26
	Tennis	1930.13	1686.55	1448.31
	Mixed martial arts	2637.12	2778.58	2515.05
	Skating	2118.00	1760.72	2090.61
	Fitness	1628.07	1782.75	1669.72
	Other sports	2150.34	1942.78	1876.89
	Not a sport	1932.30	2226.91	1963.66
Main content	χ^2^ (df)	380.56 (9)	746.10 (9)	243.63 (9)
	*p*	0.00	0.00	0.00
	Mean Rank			
	Overseas sport game analysis	1598.18	1706.66	1587.04
	Domestic sport competition analysis	1432.00	1344.19	1147.02
	Lessons and information	1432.95	1167.85	1598.99
	Specific athletes and clubs	1482.88	1829.42	1427.69
	Training	2212.53	2285.65	2081.36
	Entertainment	2075.09	2088.91	1887.21
	Daily life sharing	1878.80	2147.29	1891.83
	Eating show (mukbang)	2351.46	2754.80	2338.07
	Product advertisement	1020.18	1186.92	888.13
	Others	1244.87	1578.01	1161.32
Whether the main content matches the athlete’s event	χ^2^ (df)	62.85 (2)	217.126 (2)	64.04 (2)
	*p*	0.00	0.00	0.00
	Mean Rank			
	Same	1591.40	1541.59	1594.52
	Non-match (other event)	1859.86	1859.05	1706.90
	Content unrelated to sport	1947.87	2230.93	1973.33

**Table 7 behavsci-14-00700-t007:** Differences in content characteristics according to YouTube channels of active and retired athletes. Unit: cases (%).

		Current Player	Retired Player	Sum
Sports type	Baseball	3 (0.3)	1 (0.0)	4 (0.1)
	Football	294 (26.4)	400 (18.3)	694 (21.0)
	Basketball	2 (0.2)	128 (5.8)	130 (3.9)
	Volleyball	13 (1.2)	0 (0.0)	13 (0.4)
	Golf	209 (18.7)	1092 (49.8)	1301 (39.4)
	Boxing	8 (0.7)	126 (5.8)	134 (4.1)
	Tennis	0 (0.0)	103 (4.7)	103 (3.1)
	Mixed martial arts	122 (10.9)	87 (4.0)	209 (6.3)
	Skating	143 (12.8)	1 (0.0)	144 (4.4)
	Fitness	28 (2.5)	28 (1.3)	56 (1.7)
	Other sports	32 (2.9)	12 (0.5)	44 (1.3)
	Not applicable to sport	261 (23.4)	213 (9.7)	474 (14.3)
	χ^2^ = 859.17, df = 11, *p* < 0.001			
Main content	Overseas sport match analysis	38 (3.4)	31 (1.4)	69 (2.1)
	Domestic sport match analysis	44 (3.9)	15 (0.7)	59 (1.8)
	Lessons and information	208 (18.7)	1100 (50.2)	1308 (39.6)
	Specific athletes or clubs	258 (23.1)	379 (17.3)	637 (19.3)
	Training	99 (8.9)	114 (5.2)	213 (6.4)
	Entertainment	309 (27.7)	352 (16.1)	661 (20.0)
	Daily life sharing	75 (6.7)	104 (4.7)	179 (5.4)
	Eating show (mukbang)	33 (3.0)	28 (1.3)	61 (1.8)
	Product advertising	21 (1.9)	59 (2.7)	80 (2.4)
	Others	30 (2.7)	9 (0.4)	39 (1.2)
	χ^2^ = 373.98, df = 9, *p* < 0.001			
Whether the main content matches the athlete’s event	Same	777 (69.7)	1920 (87.6)	2697 (81.6)
	Non-match (other event)	75 (6.7)	59 (2.7)	134 (4.1)
	Content unrelated to sport	263 (23.6)	212 (9.7)	475 (14.4)
	χ^2^ = 158.36, df = 2, *p* < 0.001			
Informant	Channel operator (No other sources of information)	452 (40.5)	1218 (55.6)	1670 (50.5)
	General public	62 (5.6)	215 (9.8)	277 (8.4)
	Former sports player	52 (4.7)	227 (10.4)	279 (8.4)
	Active athletes	402 (36.1)	246 (11.2)	648 (19.6)
	Sport expert	52 (4.7)	198 (9.0)	250 (7.6)
	Celebrity	88 (7.9)	85 (3.9)	173 (5.2)
	Others	7 (0.6)	2 (0.1)	9 (0.3)
	χ^2^ = 356.11, df = 1, *p* < 0.001			

**Table 8 behavsci-14-00700-t008:** User reactions to athlete-run YouTube channel content.

	Analysis Value	Average	Standard Deviation	Minimum Value	Maximum Value
User Reaction					
Number of views		289,910.81	701,795.75	740	19,396,173
Number of comments		347.83	812.27	0	17,200
Number of likes		3937.87	9339.15	15	278,400

**Table 9 behavsci-14-00700-t009:** Differences in user reactions according to YouTube channels of retired and current athletes.

		Average Ranking	Mann–Whitney U	*p*
Number of views	Current athlete	1947.15	894,062.00	0.000
	Retired athlete	1504.06		
Number of comments	Current athlete	1955.71	884,522.00	0.000
	Retired athlete	1499.71		
Number of likes	Current athlete	1922.88	921,119.50	0.000
	Retired athlete	1516.41		

## Data Availability

The raw data supporting the conclusions of this article will be made available by the authors on request.

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
