# Peer review of "A Study on the Characteristics of Sports Athletes’ YouTube Channels and User Reactions"

_behavsci, 2024, doi:10.3390/bs14080700_

Round 1

Reviewer 1 Report

Comments and Suggestions for Authors

The paper is interesting and tackle a contemporary phenomenon on South Korea.

Text would need a final revision to eliminate what look to be a type mistake as "soccor" (line 149 page 4) and "Chai-square" (line 311 page 8). These are small things, but there is something that need be reviewed.

The data analysis just verify if some variables are impacted by others, but do not discuss the patterns of the impacts. For example, Table 4 shows the Chi-square test about the association between Video Length and Player Status (Current or Retired). The data on the table suggests Current Player uses more frequently 3-10 min videos that Retired Player, and Retired Player uses more frequently 10-20 min videos than Current Player. However, the author did not tested the statistic significance of this two affirmation. Besides, what could be said about the use of shortest (<3 min length) and longest (>20 min) videos. Are they use statically different between Player Status?

The authors showed there is an association Video Length and Player Status, but did not detailed the characteristics of the association.

This is something that authors should clarify.

Comments on the Quality of English Language

Text would need a final revision to eliminate what look to be a type mistake as "soccor" (line 149 page 4) and "Chai-square" (line 311 page 8). 

Author Response

Comments

Responses

Text would need a final revision to eliminate what look to be a type mistake as "soccor" (line 149 page 4) and "Chai-square" (line 311 page 8). These are small things, but there is something that need be reviewed.

Thank you for the helpful comments. We have reviewed and corrected the overall grammar and typos.

The data analysis just verify if some variables are impacted by others, but do not discuss the patterns of the impacts. For example, Table 4 shows the Chi-square test about the association between Video Length and Player Status (Current or Retired). The data on the table suggests Current Player uses more frequently 3-10 min videos that Retired Player, and Retired Player uses more frequently 10-20 min videos than Current Player. However, the author did not tested the statistic significance of this two affirmation. Besides, what could be said about the use of shortest (<3 min length) and longest (>20 min) videos. Are they use statically different between Player Status?

The authors showed there is an association Video Length and Player Status, but did not detailed the characteristics of the association.

We have revised and added the relevant content to the discussion section.

(Page 14, Line 420 – 434)” These results align with Cheng et al. [32], who found that more than 97.8% of sports content posted on YouTube lasted fewer than 600 seconds. The prevalence of snack culture, which involves enjoying short periods of leisure (approximately 10 minutes) while consuming snacks, has become a universal cultural phenomenon. YouTube is a representative platform where snack culture content is distributed and consumed [81]. Accordingly, it is believed that sports athletes are also producing videos that match this cultural trend. In particular, there was a notable difference in video length depending on whether the athletes were active or retired, with a higher proportion of shorter content (<10 minutes) found in the YouTube channels of current athletes. Generation Z (born in the mid-1990s to early 2000s), who are accustomed to watching videos on mobile devices, prefer short content that can be easily watched anytime and anywhere [82]. Consequently, this phenomenon is more pronounced in the YouTube content of relatively younger, current sports athletes.

However, an analysis of user reactions based on the length of content on athlete-run YouTube channels found that longer videos received relatively higher user reactions.”

Reviewer 2 Report

Comments and Suggestions for Authors

I appreciate the chance to review this essay. The study topic piqued my attention, and it was highly pertinent to the current, but the study is too old: January 1, 2020 to December 31, 2021.
The article has an intriguing title and, in my opinion, a well-structured article. Even though the abstract has to be rewritten>

A single paragraph of about 200 words maximum. For research articles, abstracts should give a pertinent overview of the work. We strongly encourage authors to use the following style of structured abstracts, but without headings: (1) Background: Place the question addressed in a broad context and highlight the purpose of the study; (2) Methods: briefly describe the main methods or treatments applied; (3) Results: summarize the article’s main findings; (4) Conclusions: indicate the main conclusions or interpretations. The abstract should be an objective representation of the article and it must not contain results that are not presented and substantiated in the main text and should not exaggerate the main conclusions.

The goal is evident, but the references may need to be revised (add new publications - in the last 3 years/ delete very old publications.

The well-defined and pertinent research question for the topic
In my opinion, the statistical analysis might be changed.  First of all, I don't understand the logic in table 4    Table 4              Current player   Retired player  Former sports player  52(4.7)       227(10.4)   Active sports player  402(36.1)     246(11.2)     Secondly, I consider Anova's analysis more appropriate. You could analyze the number of views, comments, and likes by sport type and content in the same time and try to find out if there are interactions between them.  By the way the "number of views, comments, and likes " are not characteristics of the content, but of the video.... You also could examine if the Current player/ Retired player has an interaction with the type of sport and the content published. In line 400 and in discussions you talk about correlation ("however, with respect to athlete-run YouTube content, it was found that longer video lengths correlated with higher user reactions"), but no correlation coefficient is calculated. In my opinion, the article needs major changes!

Author Response

Comments

Comments

Even though the abstract has to be rewritten>

Thank you for your valuable comments. We have revised the Abstract as follows.
“This study examined the content characteristics and user responses of athlete-run sports YouTube channels, providing empirical insights for content production strategies and contributing to the development of athlete-run sports YouTube channels. Content analysis was conducted on 3,306 videos posted on 20 popular YouTube channels of South Korean athletes from January 1, 2020, to December 31, 2021. The formal characteristics analyzed included video length, the presence of foreign language subtitles, paid advertisements, and information sources. The content characteristics examined were the types of sports events, main content themes, and whether the content matched the athlete's sport. Results revealed significant differences in content characteristics and user responses based on whether the athletes were active or retired. This study's distinctive contribution lies in highlighting the evolving role of athletes as content creators and providing strategic implications for enhancing the competitiveness of athlete-run sports YouTube channels. Future research should consider a broader range of sports YouTubers and a wider variety of YouTube channels to gain comprehensive insights into the sports content ecosystem on this platform.”

The goal is evident, but the references may need to be revised (add new publications - in the last 3 years/ delete very old publications.

Based on the reviewer's valuable comments, we have added various up-to-date references. (See page 18 – 22)

In my opinion, the statistical analysis might be changed.  First of all, I don't understand the logic in table 4    Table 4              Current player   Retired player  Former sports player  52(4.7)       227(10.4)   Active sports player  402(36.1)     246(11.2)

The mixed usage of terms such as Retired/Former and Active/Current has caused confusion among readers. Therefore, we have standardized these terms to Retired and Active in Table 4.

Secondly, I consider Anova's analysis more appropriate

ANOVA analysis requires the dependent variable to be at an interval or ratio scale. Since the dependent variable is at a nominal scale, we conducted a Chi-square test instead.

We have also added to the research methodology that non-parametric statistical analysis was conducted due to the dependent variable not satisfying normality.

(Page 7, line 295 – 301) “As a result of the normality test conducted on user responses, which are the dependent variables of this study, it was found that the data did not satisfy normality. When data do not meet normality, non-parametric statistical analysis is performed [77, 78, 76]. The Mann-Whitney U Test is a non-parametric statistical technique that compares the ranks of data included in two groups, and the Kruskal-Wallis Test compares the ranks of data included in three or more groups to identify differences between groups [79, 80].”

(Page 9, line 337 – 340) ” This method is appropriate because it is used to test the independence of categorical variables, allowing for the comparison of the distribution of formal content characteristics across the two distinct groups of active and retired YouTube channel operators.”

You could analyze the number of views, comments, and likes by sport type and content in the same time and try to find out if there are interactions between them

The research results present the differences in user responses according to the formal characteristics of sports athletes' YouTube channel content in section 3.2 (Page 8), and the differences in user responses according to the content characteristics of sports athletes' YouTube channel content in section 3.5 (Page 10).

By the way the "number of views, comments, and likes " are not characteristics of the content,

In Table 1, views, comments, and likes are presented as user reaction. (Page 6)

You also could examine if the Current player/ Retired player has an interaction with the type of sport and the content published. In line 400 and in discussions you talk about correlation ("however, with respect to athlete-run YouTube content, it was found that longer video lengths correlated with higher user reactions"), but no correlation coefficient is calculated. In my opinion, the article needs major changes!

Thank you for the helpful comments. This was a mistake on our part while writing. We have revised the sentence as follows to reduce any potential confusion.

(Page 14, line 433) “an analysis of user reactions based on the length of content on athlete-run YouTube channels found that longer videos received relatively higher user reactions.”

Reviewer 3 Report

Comments and Suggestions for Authors

This is a very interesting and worthwhile paper which analyses the YouTube channels of active and retired Korean athletes. It provides an analysis of differences between the channels of the ‘active’ and ‘retired’ content creators, paying particular attention to video length, viewer reactions, captions, and advertisements, as well as an analysis of channel content. The paper makes a useful contribution since sports-related channels have seen a huge increase on YouTube, with athletes now adopting YouTubing as a way of interacting with their fans.

Much of the paper is strong and well-written. My comments on many sections are very light. The exception is the Literature Review section, which I thought was weak in some respects.

In my view, it would be useful for the authors to address the following points:

·         The Abstract should more explicitly state the distinctive contribution made by this paper to the academic literature.

·         The Introduction should make clear which area of literature this paper contributes to.

·         The research questions presented on page 3 need more justification. It should be explained in more detail why these points of focus have been chosen. The research questions should also be referred to more explicitly later in the paper, when reporting the Results. It seems implied how the Results relate to the questions, but this could be stated explicitly and clearly.

·         The literature review section should start by elaborating why some specific areas of literature have been chosen for review—in other words, by justifying the *scope* of the review.

·         It is not clear to me why section 2.1 focusses on Korea specifically. Surely it would be better to address literature on YouTube sports content more generally? This way we could see how this paper makes a contribution to international scholarship. In other words, just because the empirical part of this paper focusses on Korean YouTube channels does not mean that the literature review should also adopt this national perspective.

·         Both literature review sections (2.1 and 2.2) are weak. They describe the underlying phenomenon, rather than reviewing the literature. We need to know what the literature argues, what the literature emphasises, what the literature analyses, etc. We then need to know what the authors’ criticisms of these existing aspects of the existing literature are. We don’t need a guide to recent developments on YouTube; that’s not the purpose of a literature review section.

·         It would be useful, if possible, to indicate how the coding categories (summarised in Table 1) correspond to the research questions.

·         In section 4, where specific statistical tests are used, these should be briefly justified (in many cases, a single sentence should be adequate).

·         The discussion section does a good job of synthesising the findings, but it does not really establish how these findings contribute to the literature. Once the literature review has been reformulated (see above), this ought to be easier to do.

·         The Conclusion section should more explicitly consider the limitations of the present study.

Author Response

Comments

Comments

The Abstract should more explicitly state the distinctive contribution made by this paper to the academic literature.

Thank you for your valuable comments. We have revised the Abstract as follows.
“This study examined the content characteristics and user responses of athlete-run sports YouTube channels, providing empirical insights for content production strategies and contributing to the development of athlete-run sports YouTube channels. Content analysis was conducted on 3,306 videos posted on 20 popular YouTube channels of South Korean athletes from January 1, 2020, to December 31, 2021. The formal characteristics analyzed included video length, the presence of foreign language subtitles, paid advertisements, and information sources. The content characteristics examined were the types of sports events, main content themes, and whether the content matched the athlete's sport. Results revealed significant differences in content characteristics and user responses based on whether the athletes were active or retired. This study's distinctive contribution lies in highlighting the evolving role of athletes as content creators and providing strategic implications for enhancing the competitiveness of athlete-run sports YouTube channels. Future research should consider a broader range of sports YouTubers and a wider variety of YouTube channels to gain comprehensive insights into the sports content ecosystem on this platform.”

The Introduction should make clear which area of literature this paper contributes to.

Thank you for the valuable comments. We have clearly outlined the contribution of this paper to the field.
(Page 2, line 42-43) “This study contributes to the literature on digital media, social media communication, and sports communication by examining athlete-run sports YouTube channels.”

The research questions presented on page 3 need more justification. It should be explained in more detail why these points of focus have been chosen. The research questions should also be referred to more explicitly later in the paper, when reporting the Results. It seems implied how the Results relate to the questions, but this could be stated explicitly and clearly.

In response to the editor's comments, we have replaced our RQs with two comprehensive questions. Based on the reviewer's valuable comments, we have provided justification for the new RQs.

(Page 4, line 190 – 221) “In response, this study aims to examine the characteristics and user reactions to the content of sports YouTube channels operated by active and retired athletes. Specifically, this research seeks to address the following guiding questions: (1) What are the key characteristics and formats of the content produced on athlete-run sports YouTube channels? (2) How do user reactions vary based on these content characteristics and the athlete’s status (active vs. retired)? This comprehensive examination provides insights into the unique attributes of athlete-run sports YouTube content and offers strategic implications for enhancing the competitiveness of these channels, thus holding both academic and industrial significance.

By addressing these guiding questions, the study aims to offer a comprehensive understanding of the dynamics of athlete-run sports YouTube channels. Specifically, examining the key characteristics and formats of these channels is essential for identifying common practices and industry standards, enabling comparisons with other types of sports content on YouTube and determining elements contributing to success. Investigating how user reactions vary based on content characteristics and the athlete’s status provides insights into audience preferences and engagement, crucial for optimizing videos for user satisfaction and higher engagement rates. Understanding whether content characteristics differ between active and retired athletes can help tailor content strategies to the unique needs and capabilities of each group, given their different motivations and resources.

Exploring these aspects sheds light on the types of content that attract and retain viewers, offering a deeper understanding of what makes certain content popular. Analyzing differences in user responses based on content characteristics allows content creators to identify which types of content are more effective in engaging audiences and fostering loyalty. Understanding if and how content characteristics differ between active and retired athletes can provide insights into how an athlete’s status influences their content creation and audience engagement strategies.

Assessing user reactions to the content on athlete-run sports YouTube channels is vital for evaluating the success and impact of these channels, helping to understand audience sentiment and overall content reception. Collectively, these inquiries aim to offer strategic implications for enhancing the competitiveness and effectiveness of athlete-run sports YouTube channels, thus holding both academic and industrial significance.”

The literature review section should start by elaborating why some specific areas of literature have been chosen for review—in other words, by justifying the *scope* of the review.

In response to the editor and reviewer's comments, we have combined and rewritten the literature review and introduction sections.
(Page 1 – 5, line 25 – 222)

It is not clear to me why section 2.1 focusses on Korea specifically. Surely it would be better to address literature on YouTube sports content more generally? This way we could see how this paper makes a contribution to international scholarship. In other words, just because the empirical part of this paper focusses on Korean YouTube channels does not mean that the literature review should also adopt this national perspective.

In response to the reviewer's comments, we have revised the literature review to include more general content, ensuring a thorough review.

(Page 2 – 4, line 74 – 189).

Both literature review sections (2.1 and 2.2) are weak. They describe the underlying phenomenon, rather than reviewing the literature. We need to know what the literature argues, what the literature emphasises, what the literature analyses, etc. We then need to know what the authors’ criticisms of these existing aspects of the existing literature are. We don’t need a guide to recent developments on YouTube; that’s not the purpose of a literature review section.

It would be useful, if possible, to indicate how the coding categories (summarised in Table 1) correspond to the research questions.

Thank you for the helpful comments. In response to the editor's feedback, we have replaced the RQs with more comprehensive questions, and all the content in Table 1 now encompasses these overarching RQs.

In section 4, where specific statistical tests are used, these should be briefly justified (in many cases, a single sentence should be adequate)

Thank you for the insightful comments. In each section where statistical analysis was conducted, we have briefly explained the reasons for using the respective analysis methods.

(Page 8, line 323 – 326) “These non-parametric methods are appropriate because they do not assume a normal distribution of the data and are suitable for comparing differences between groups with ordinal or non-normally distributed interval data, which is common in user response metrics on social media platforms.”

(Page 9, line 336 – 339) “This method is appropriate because it is used to test the independence of categorical variables, allowing for the comparison of the distribution of formal content characteristics across the two distinct groups of active and retired YouTube channel operators.”

(Page 11, line 372 – 375) “This method is appropriate because it is a non-parametric test that allows for the comparison of more than two independent groups, making it suitable for analyzing user responses that are likely ordinal or non-normally distributed across different content characteristics.”

The discussion section does a good job of synthesising the findings, but it does not really establish how these findings contribute to the literature. Once the literature review has been reformulated (see above), this ought to be easier to do.

Thank you for the insightful comments. In the discussion chapter, we have elaborated on how our findings contribute academically.

(Page 16 – 17, line 569 – 603)

The Conclusion section should more explicitly consider the limitations of the present study.

Thank you for your valuable comments. Based on the reviewer's suggestions, we have completely rewritten the Conclusion chapter.

(Page 17, line 605 – 639)

Round 2

Reviewer 1 Report

Comments and Suggestions for Authors

I am satisfied with the new version of the paper

Reviewer 2 Report

Comments and Suggestions for Authors

Thank you for the answer

I really appreciate the improvements brought to the article.